# Molecular Epidemiology of the Norwegian SARS-CoV-2 Delta Lineage AY.63

**DOI:** 10.3390/v14122734

**Published:** 2022-12-07

**Authors:** Line Victoria Moen, Hilde Synnøve Vollan, Jon Bråte, Olav Hungnes, Karoline Bragstad

**Affiliations:** Division of Infectious Disease Control and Environmental Health, Norwegian Institute of Public Health, 0213 Oslo, Norway

**Keywords:** Delta, AY.63, Norway, SARS-CoV-2

## Abstract

Extensive genomic surveillance has given great insights into the evolution of the SARS-CoV-2 virus and emerging variants. During the summer months of 2021, Norway was dominated by the Pango lineage AY.63 which is a sub-lineage of the highly transmissible Delta variant. Strikingly, AY.63 did not spread in other countries to any significant extent. AY.63 carried a key mutation, A222V, in the spike protein, as well as the deletion of three residues in nsp1. Although these mutations are close to functionally important areas, we did not find any evidence that they induced higher fitness compared to other Delta lineages. This variant was introduced to Norway at a time when there were low levels of SARS-CoV-2 and contact-reducing measures were relaxed, which probably explains why the lineage rose so quickly. Furthermore, we found that the lack of imports of AY.63 from other countries probably led to the eventual demise of the lineage in Norway.

## 1. Introduction

Delta was declared a Variant of Concern (VOC) by the World Health organization in May 2021 [1]. One of the Delta sub-lineages of SARS-CoV-2, designated with the Pango nomenclature AY.63 (alias of B.1.617.2.63) [2], was nicknamed “the Norwegian lineage” because it was almost exclusively found in Norway.

Genomic surveillance using whole virus genome sequences has been an important tool to observe and study the evolution and spread of SARS-CoV-2, and by October 2022, more than 13 million partial or complete genomes have been submitted to the GISAID (Global Initiative on Sharing Avian flu Data) EpiCoV database [3]. Throughout the first months of the pandemic, the virus evolved slowly, but this changed by the end of 2020 when a continuous rise and displacement of regionally restricted variants was seen [4,5,6]. Some of these new variants were labelled VOCs due to the risk of more severe disease or increased transmissibility [7].

The SARS-CoV-2 genome encodes 29 proteins, including four structural proteins (spike, envelope, membrane, and nucleocapsid) and 16 non-structural proteins (nsp) [8]. The spike protein consists of two subunits called S1 and S2 [9]. S1 harbors the important receptor binding domain (RBD) (residues 319 to 541) which directly interacts with ACE-2, as well as the N-terminal domain (NTD) (residues 14-305), which is poorly characterized but thought to be important for cell entry [10]. The S2 domain is involved in the membrane fusion process.

The Delta VOC, designated B.1.617.2, was first detected in October 2020 in India [1]. Delta has been shown to be 50% more transmissible than Alpha [11] and by July 2021, Delta was the dominant SARS-CoV-2 lineage in almost all continents [12].

Delta carried ten mutations in the spike protein compared to the original Wuhan strain [13]; two in the RBD (L452R and T478K), five in the N-terminal domain (T19R, G142D, E156G, Δ157-158), two near the furin cleavage site (D614G and P681R), and one mutation in the S2 region (D950N) [14]. The Delta lineage quickly diversified into over 200 sub-lineages, all denoted with AY.X [2]. 

Before the emergence of the VOCs, the pandemic in Norway was characterized by an influx of new variants that were quickly eradicated due to strict regulations that minimized social gatherings and close contact [15]. When the first Delta lineages appeared in Norway in April 2021, the number of SARS-CoV-2 cases was therefore relatively few, and the first lineages soon disappeared. Nevertheless, Delta became the dominant lineage by July 2021 [16]. 

The Delta lineage AY.63 had a massive and abrupt increase in Norway during the summer of 2021. Notably, AY.63 was almost exclusively found in Norway. It is unclear why AY.63 spread quickly throughout Norway but did not spread to the same extent in other countries. AY.63 had the additional spike A222V mutation, also seen in the B.1.177 lineage in Europe the year before [17]. Furthermore, AY.63 had a deletion of residues 141–143 in the nsp1 protein (encoded by ORF1a). As AY.63 likely descended from a single or very few import(s), and mainly spread in Norway, it provides a unique opportunity to study the evolution and within-country spread of this variant. Even though AY.63 for a while became dominant in Norway, it eventually disappeared and was replaced by other Delta lineages. Interestingly, these were lineages that had been co-circulating with AY.63, and some were even introduced earlier than AY.63. Why these lineages did not displace AY.63 earlier remains unclear. 

In the present study, we have characterized the introduction and spread of AY.63 in Norway, as well as comparing the molecular characteristics against co-circulating Delta lineages.

## 2. Materials and Methods

### 2.1. Sequence Data and Metadata Collection

As part of COVID-19 surveillance, the national reference laboratory for emerging coronaviruses at the Norwegian Institute of Public Health (NIPH) solicits virus-positive samples for sequencing from the regional microbiology laboratories, as well as sequences produced by these laboratories. Most sequences of good quality (>94% coverage) are shared with GISAID by NIPH. All sequence data used in this study for comparative analyses were retrieved from the GISAID database16 on 10 October 2022.

### 2.2. Phylogenetic Analysis

Phylogenetic analyses of Delta sequences were conducted using NextStrain [18] and its specific pipeline for SARS-CoV-2 (https://github.com/nextstrain/ncov; accessed on 10 October 2022). The data were divided into two sets, one consisting of all sequences with the AY.63 Pango lineage annotation (AY.63 dataset), and one with all other Delta sequences (i.e., with either the Pango lineage annotation “B.1.617.2” or beginning with “AY.”) collected between 01 March 2021, and 31 December 2021, and AY.63 removed (Delta dataset). To visualize AY.63 in the context of all Delta strains, the NextStrain analysis was set to include all AY.63 sequences and to subsample evenly across all other lineages. To focus on AY.63, the NextStrain analysis was set to include all sequences from the AY.63 dataset and to sample only a small selection of the most closely related Delta sequences to provide a reasonable outgroup. The complete set of GISAID IDs used in the analysis presented in Figure 2 is available via the GISAID Epi Set identifier EPI_SET_221012yt (doi: 10.55876/gis8.221012yt).

### 2.3. Plotting and Visualization

The plots showing the geographic distribution of AY.63 in Norway, the cumulative number of cases, and the average SNP frequencies were generated using R [19] packages tidyverse [20] and ggplot2 [21]. The area chart showing the relative distribution of the various Pango lineage lineages in Norway was created using Microsoft Excel. A Venn-diagram of spike and ORF1a mutations shared between the Delta variants B.1.617.2, AY.43, AY.63, and AY.4 was adapted from a query-search on CoV-Spectrum (Norway; All samples; week 23–48, 2021) [22].

### 2.4. Pairwise SNP Distance and Import Analysis

To calculate pairwise SNP distances, sequence alignments were generated for each of the lineages B.617.2, AY.63, AY.4, AY.43, AY.122, and AY.127 using NextStrain [18] on the same dataset as described above. The final alignments were converted into a matrix of pairwise SNP distances using the R packages pairsnp (https://github.com/gtonkinhill/pairsnp; accessed on 10 October 2022) and harrietr (https://github.com/andersgs/harrietr; accessed on 10 October 2022). The number of imports into Norway for each of the Delta lineages was estimated using the R package LineageHomology (https://github.com/magnusnosnes/LineageHomology; accessed on 10 October 2022) [15]. As inputs for the analysis, the phylogenetic trees and metadata files generated by NextStrain were used. For all lineages except AY.63, we assumed that the rate of imports into Norway vs. local transmissions was 10, and for AY.63, we assumed a rate of 0.5.

### 2.5. Protein Structure Visualization

The SARS-CoV-2 modeled spike protein structure by Cao et al. [23] (available at https://charmm-gui.org/?doc=archive&lib=covid19; accessed on 11 November 2022) was cleaned using YASARA/WHATIF Structure Twinset v20.3.4 [24,25]. This model is based on the spike model with the Protein Data Bank (PDB) identifier 6VXX [26] using all-atom molecular dynamics simulation closing any gaps in the original solved structure. The AY.63 mutations were colored using r3dmol [27]. The A222V mutation in Figure 6B was visualized using YASARA/WHATIF Structure Twinset using the swap residue function.

## 3. Results and Discussion

### 3.1. Introduction and Spread of AY.63 in Norway

The Delta variant was repeatedly introduced to Norway, mainly by B.1.617.2 and AY.122 variants. Nevertheless, AY.63 became the dominant lineage (Figure 1A) by the end of June 2021. The first case of AY.63 in GISAID was registered in Norway on 10 June 2021 (GISAID ID: EPI_ISL_2673741). After the initial observation, the lineage was rapidly detected in 9 out of 11 counties, with almost 300 sequenced samples within the next three weeks classified as AY.63 (Figure 1B). At this time, 30–50% of all SARS-CoV-2 cases were sequenced in Norway [28]. By week 27–28, AY.63 had a 49 % prevalence in Norway and reached 1000 cases within 53 days. In comparison, the other most common variants circulating reached 1000 cases only after 75 days (AY.4) and 103 days (AY.43) (Figure 1C). AY.63 continued to increase steadily until the beginning of October 2021 when more than 2000 samples of AY.63 had been detected (Figure 1C). 

AY.63 mainly spread in the southern and southeastern parts of Norway before it migrated to the western and northern regions (Figure 1B and Figure 2). It is also evident from the phylogenetic analysis that many nearly identical strains were dispersed across the country, supporting a rapid initial spread of AY.63 (Figure 2). After the initial spread across the country, several regional outbreaks occurred, particularly in the Viken and “Troms og Finnmark” counties. 

In Norway, the first months of 2021 were characterized by strong social restrictions, increasing levels of immunity, and higher vaccination rates which led to a decline of the Alpha variant (Figure 1A). The Delta variant, which was more transmissible [11,29] superseded Alpha and by July 2021, it became the dominant lineage in Norway. At that time there were still some contact-reducing regulations in place in Norway, such as using facemasks and maintaining a two-meter distance from other people in public spaces, but the regulations eased as the vaccination rate went up. The introduction of AY.63 into Norway therefore coincided with low overall levels of SARS-CoV-2 and a relaxation of contact-reducing regulations. This, together with the fact that Delta was highly transmissible, probably led to the massive increase in AY.63. It seemed like AY.63 would be the dominate lineage in Norway, and apparently it had an advantage over B.1.617.2 (parent Delta lineage) (Figure 1A). Nevertheless, the number of AY.63 cases decreased with diminishing numbers being observed until week 48, when AY.63 apparently became extinct in Norway.

### 3.2. Origin, Spread, and Extinction of AY.63 in Europe

AY.63 formed a distinct clade in the phylogeny of all Delta sequences (Figure 3A). AY.63 has several characteristic mutations, most notably the combination of A222V in the spike protein and the 141–143 deletion in nsp1 (encoded by the ORF1a gene). Both genomic signatures have been observed in other Delta lineages as well. The A222V mutation probably has a deep origin in Delta, close to the split between the two major groups (21I and 21J according to the NextClade nomenclature [30]), while the deletion in nsp1 is less common but has been observed several times in the 21I clade as well (Figure 3A).

Interestingly, AY.63 was almost exclusively found in Norway (2570 cases) compared to the rest of the world (215 cases). Besides Norway, AY.63 was most frequently observed in the two neighboring countries Denmark (168 cases) and Sweden (26 cases) (Figure 3B). Thus, this lineage was clearly much more common in Norway than in the other European countries. The Nordic countries overall had similar regulations in place to limit the spread of SARS-CoV-2 (especially in Norway and Denmark) [31]. Hence, different governmental policies are likely not to be the main reason for the dominance of AY.63 in Norway. In fact, both Denmark and Sweden had similar trajectories of COVID-19 cases as Norway during the summer and autumn of 2021, but with other dominating Delta lineages [32,33]. The differences in lineage composition were probably the result of founder effects due to an overall rise in cases from very low numbers. A few cases of AY.63 were also exported from Norway without resulting in any large outbreaks (Figure 2). This supports the hypothesis that the dominance of this lineage in Norway does not necessarily indicate the higher transmissibility of AY.63 as compared to the other Delta lineages.

Although the first published sequenced samples of AY.63 were registered in Norway, the phylogenetic analysis suggests that the lineage originated outside of the country. However, the “Norwegian cluster” and the “European cluster” (which consists predominantly of Danish samples) are genetically distinct and separated by relatively long branches (Figure 2). It is therefore possible that the positions of the basal clusters in the AY.63 phylogeny are affected by branching artifacts due to homoplasies in the two groups. In addition, the earliest registered AY.63 sequence (Norway GISAID ID: EPI_ISL_2673741) lacks the 141–43 deletion in nsp1 and clusters on a separate branch outside of the other Norwegian strains.

One interpretation of the phylogenetic analysis in Figure 2 is that the Norwegian AY.63 was imported from another European country, most likely Denmark. Of the non-Norwegian countries, Denmark had by far the greatest number of AY.63 samples (Figure 3), hence the “European cluster” was mostly composed of Danish sequences. However, the samples in this cluster are from later dates than the Norwegian cluster (Figure 2). The genomic surveillance of SARS-CoV-2 in Denmark has been exceptionally comprehensive, as more than 90% of all positive cases were sequenced in the period that AY.63 circulated [34,35]. It is therefore unlikely that Danish samples closely related to the Norwegian ones would have gone undetected. Based on the large genetic distance between the Norwegian and the European clusters it is more likely that there existed a large unsampled diversity of AY.63 prior to the first observed sample in Norway, and that when and where AY.63 originated remains unknown.

Given the genetic distinctness of the Norwegian AY.63 sequences, it is highly likely that the entire diversity of AY.63 in Norway arose from a single or very few imports, with a continued evolution within Norway. This is corroborated by our import analysis, which estimated three imports of AY.63 into Norway (Figure 4). Even though this number should be treated with caution as the analysis is highly sensitive to the assumed rate of imports vs. local transmissions, the number of imports is nevertheless very low compared to the estimates for the other co-circulating Delta lineages at the time (ranging from 42–254 estimated imports).

In addition to AY.63, there were also other Delta lineages circulating in Norway during the latter half of 2021. The five most common lineages (in addition to AY.63) were B.1.617.2, AY.4, AY.43, AY.127, and AY.122. These lineages seemed to have had a much lower, albeit more stable, presence than AY.63 (Figure 1A). In addition, while AY.63 died out in Norway only 174 days after it was first reported, these other lineages persisted.

One likely reason is that AY.63 was not repeatedly imported into Norway from other countries. Norway is a sparsely populated country, and imports from other countries have a huge effect on the lineage dynamics of SARS-CoV-2 [15]. As AY.63 was almost exclusively found in Norway, imports were much less likely than for the more widely circulating lineages. The lack of imports was also evident on the genetic diversity of the lineage in Norway (Figure 4). AY.63 had on average a lower genetic diversity (measured as pairwise SNPs between sequences) compared to the other lineages, despite a continuous spread across the country. It is therefore natural to assume that AY.63 disappeared in Norway because there was no replenishment from abroad and its domestic circulation was not sustainable. This latter factor may well have also been the case for other Delta lineages at the time. A similar observation was made of another Delta variant, AY.4.2.1, that also had the A222V mutation and mainly circulated in Bulgaria [36].

### 3.3. Molecular Characteristics of AY.63

As described above, AY.63 shares most of the common Delta mutations, but the combination of the two mutations is characteristic of the lineage: A222V in the spike protein and the deletion of residues 141–143 in nsp1 (Figure 5).

The A222V mutation in the spike protein has been observed in several SARS-CoV-2 lineages, including Delta (Figure 6A) [17,37,38]. Notably, A222V was present in a large cluster, B.1.177, that spread rapidly throughout Europe during the spring of 2020 [24]. No evidence was found that the A222V mutation altered the conformation of the spike protein or affected the viral entry into host cells. The wave of B.1.177 instead seemed to have been driven mostly by cross-border travel and multiple imports.

Position 222 in the spike protein is located in the NTD. There are some indications that the NTD (circled in Figure 6A) contains certain antigenically important loop structures [39] and changes there may promote immune escape. Residue 222 lies in a loop facing beta-strands (Figure 6B). Our investigation into the structural effects of having an alanine (A) or a valine (V) in position 222 shows that the conformational change is probably very small. It is, however, possible that the slightly larger side chain could create a larger steric hindrance (Figure 6B). This is supported by a neutralization experiment performed on the Delta lineage AY.4.2 that also harbors the A222V spike mutation [40]. The results showed a slightly reduced, although not significantly, sensitivity to sera from vaccinated individuals [40]. Although alanine and valine are both hydrophobic, it might affect the interaction with the immune system. It is interesting to note that the A222V mutation has emerged independently in several lineages, but it still causes no obvious changes to the spike structure or has any other notable effects.

In addition to the spike mutation A222V, AY.63 is also defined by the deletion of residues 141–143 in ORF1a. This deletion is in nsp1, an essential virulence factor that has been implicated in prolonging the infection. Nsp1 is a leader protein that inhibits the translation of host mRNA by binding to the ribosome and thereby promotes host mRNA degradation. Hence, nsp1 has been suggested as a potential antiviral drug target.

The protein consists of 180 amino acids and the deletion is located in one of the sheets between the N- and C-terminal domains (NTD; residues 10–127 and CTD; residues 148–180) [41]. Although it is not clear how the deletion affects the protein structure, deletions in nsp1 have been implicated in increased structural stability that could inhibit host translation better than the wild-type protein [42].

### 3.4. Concluding Remark

At the time of its emergence, the Delta variant rapidly outcompeted other concurrent variants due to its superior fitness. Whereas the AY.63 lineage carries some signature mutations that may possibly convey further advantageous traits to the virus, we cannot conclude that this was the reason for the relative success of this lineage in Norway during some months in 2021. The main feature of the lineage in our analysis is that it lacked an active source of reintroductions from abroad, and this thus provides an example that illustrates how, during the period public health measures still being active, domestic transmission alone could not sustain the long-term circulation of pre-Omicron SARS-CoV-2 in Norway.

## Figures and Tables

**Figure 1 viruses-14-02734-f001:**
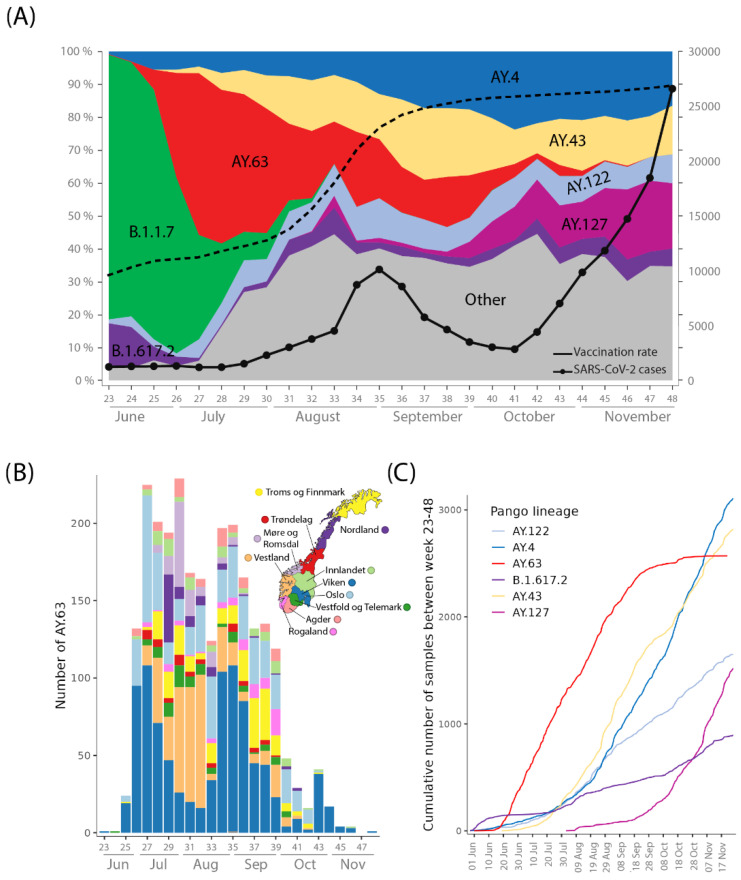
Spread of AY.63 in Norway, 2021. (**A**) Proportions of the B.1.1.7, AY.63, and the five most common Pango lineages circulating in Norway between weeks 23–48, 2021. “Other” lineages (colored grey) contain all other circulating lineages (including a few Omicron cases). The black dashed line shows the percent of people (<18 years) who had received two doses of vaccination against SARS-CoV-2. The black dotted line shows the numbers of positive SARS-CoV-2 cases in Norway (plotted on the secondary *y*-axis). Week numbers and months of 2021 are shown on the *x*-axis. (**B**) Number of sequenced samples of AY.63 colored by Norwegian county. The *X*-axis is the same as in figure (**A**). The map was adapted from Wikipedia.org (accessed on 10 October 2022). (**C**) Cumulative cases of sequenced samples of AY.63 and the five other most frequent Delta lineages in weeks 23–48.

**Figure 2 viruses-14-02734-f002:**
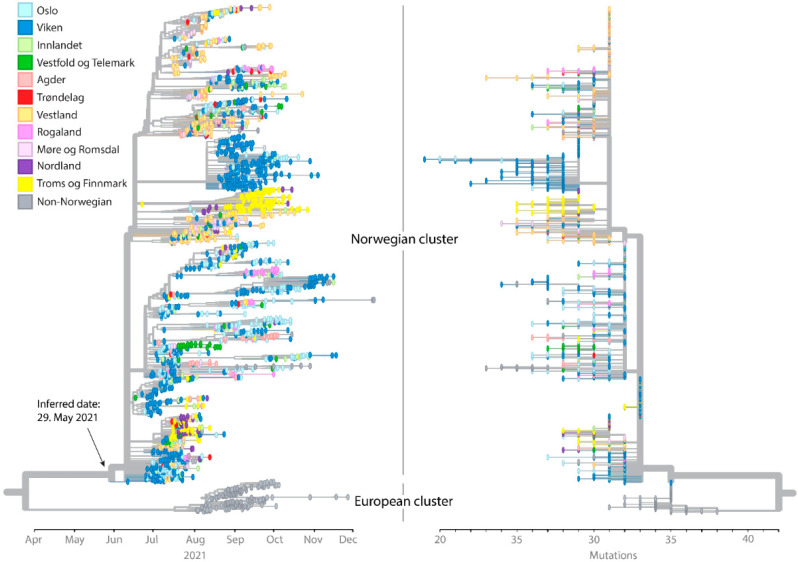
Phylogeny of AY.63. Phylogenetic analysis of AY.63 with a selection of the closest non-AY.63 Delta sequences as the outgroup (not shown). The tree on the left has the branch tips arranged according to sample date, with the internal nodes positioned on their inferred dates according to TreeTime; while on the right, the branch tips are arranged according to the genetic divergence from the root of the tree. Norwegian samples are colored according to region, and the non-Norwegian samples are indicated with grey circles.

**Figure 3 viruses-14-02734-f003:**
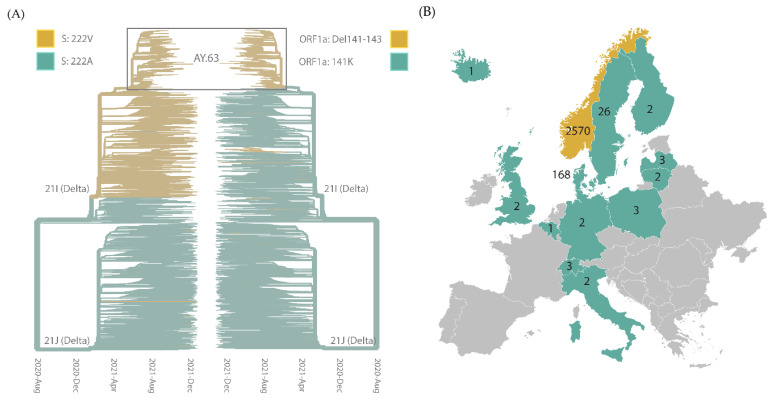
Phylogeny of the Delta lineage. (**A**) Phylogenetic analysis of Delta with the Pango lineage AY.63 and NextClade clade annotations indicated. The strains carrying the characteristic genetic signatures for AY.63, spike A222V (left tree), and ORF1a 141–143 deletions (right tree) are colored yellow. (**B**) The number of sequences with Pango lineage AY.63 published in GISAID drawn on their country of origin. In addition to the countries visualized here, there is also one sequence sampled from the USA published in GISAID. The map was adapted from Wikipedia.org, accessed on 10 October 2022.

**Figure 4 viruses-14-02734-f004:**
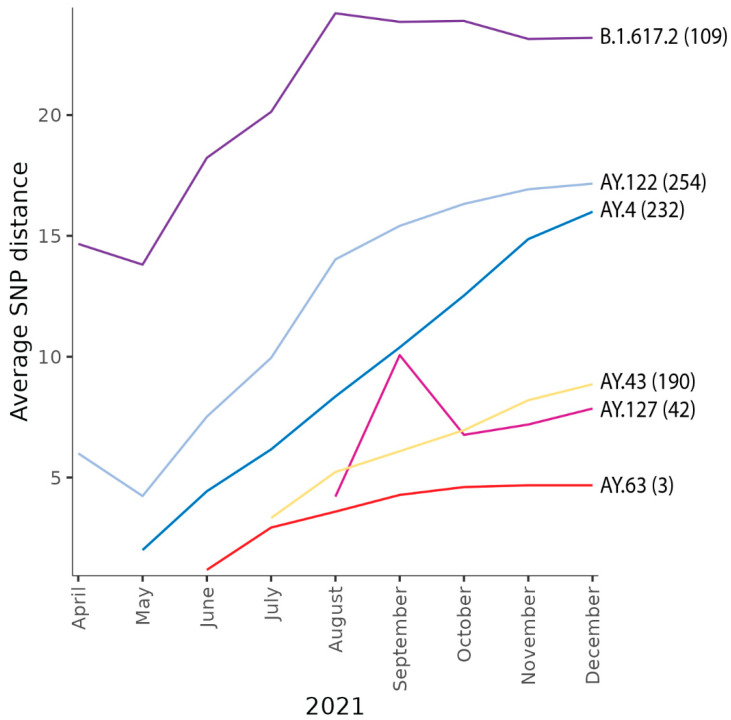
Lineage dynamics. Cumulative average pairwise SNP distances plotted over time for the Norwegian AY.63 sequences and the most frequent co-circulating lineages. To the right of each graph is the Pango lineage nomenclature for each of the lineages indicated with the estimated number of imports into Norway in parenthesis.

**Figure 5 viruses-14-02734-f005:**
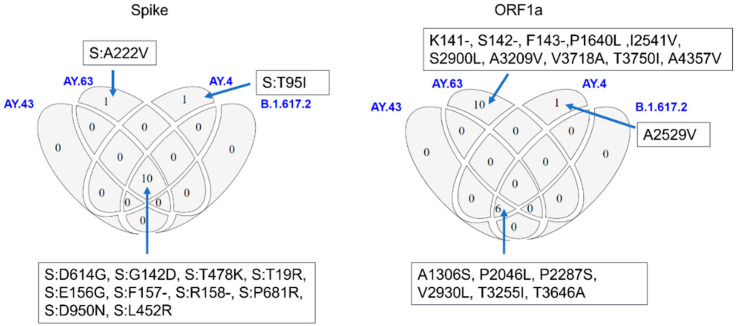
Genetic characteristics of AY.63. Overview of spike and ORF1a mutations shared between the most common Delta variants in week 23–48.

**Figure 6 viruses-14-02734-f006:**
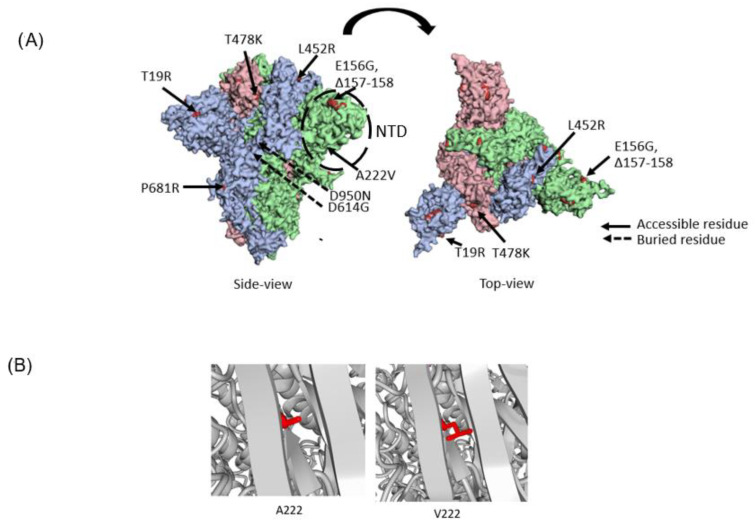
Structure analysis of AY.63 spike mutations. (**A**) Top and side-view of the spike protein where each subunit is colored differently (in the homotrimeric structure) with the mutations for AY.63 highlighted. (**B**) The A222V mutation visualized where alanine residue is to the left and valine on the right side.

## Data Availability

All data are downloadable from GISAD. The complete set of GISAID IDs used in the analysis is available via the GISAID Epi Set identifier EPI_SET_221012yt (doi:10.55876/gis8.221012yt).

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
