# Peer review of "Molecular Epidemiology of the Norwegian SARS-CoV-2 Delta Lineage AY.63"

_viruses, 2022, doi:10.3390/v14122734_

Round 1

Reviewer 1 Report

In this interesting manuscript, the authors investigate the unusual molecular epidemiology of a SARS-CoV-2 Delta variant called AY.63, which exhibited a rapidly ascending, but relatively short-lived dominance in Norway, and remarkably essentially only in Norway, before quickly dying off.  The variant carried a relatively unique combination of mutations, including A222V in the spike protein and a deletion of three residues in the non-structural protein NSP1.  It is thought that the variant arose at a time when the country was experiencing low levels of SARS-CoV-2 and a relaxation of protective guidelines.  Although there were a moderate number of cases in Denmark (168) and to a lesser extent Sweden (26), the variant never took a stronghold in those countries anywhere near that in Norway.  It is concluded that in Norway AY.63 had no replicative advantage and died off because there was not continued import of the virus from other countries.

This is considered a thorough and informative evaluation of the progression of AY.63.  The authors have done a good job both presenting and interpreting the data.  There are some relatively minor points, which addressed may enhance the manuscript.

1)    In the text, most often time points are indicated by month.  However, in Figure 1, it is labeled as week number or days since June 10, 2021.  Why not just use the date here?  As presented, it is inconvenient and distracting for the reader.

2)    Since there were relatively few cases found in Denmark and especially Sweden, is there any information available as to whether these individuals had traveled to Norway and possibly brought it back to their country?  Alternatively, could one of the infected Danes or Swedes be responsible for introducing it into Norway?  Do the dates of infection of these individuals align with either of these possibilities?

3)    With respect to the lack of an AY.63 outbreak in Denmark and Sweden, the question arises whether these two countries were or were not under relaxed protocols similar to those in place in Norway when the AY.63 outbreak arose.  If not, this may account for their comparatively lower prevalence in those two countries.  If so, this would argue against the reduced protection being responsible or the rapid outbreak of AY.63 in Norway.

4)    Line 132-4:  Here, it is discussed in reference to the data in Fig. 1B and 2 that AY.63 initially spread mainly in the southern and southeastern regions of the country.  For the convenience of the reader, the authors should indicate the geographical location of the various provinces, perhaps by the addition of a small map to this figure.

5)    Line 172: To what figure does Figure A refer to?

6)    The authors conclude that AY.63 disappeared because it was not replenished from elsewhere.  But, from where?  And why didn’t it become dominant there?

Reviewer 2 Report

Revision manuscript “Molecular Epidemiology of the Norwegian SARS-CoV-2 Delta 2 lineage AY.63”

The article provides an interesting molecular epidemiological picture of the AY.63 lineage that appeared in Norway in 2021 and die out after several weeks. The mutations detected in this lineage apparently did not provide any selective advantage. The paper provides new insights into the evolution of this virus.

Introduction:

Line 26: delete extra “has been”

Line 58, correct “linage”

Figure 2: improve the resolution

Paragraph 3.2: Figure 2A should be Figure 3A. Figure B should be Figure 3B

Based on the figure, it is impossible to distinguish the Norvegian and Danish clusters as stated in the text. Add an image that highlights this aspect.
